# Feasibility and reliability of decentralized HIV-1 viral load monitoring on self-sampled blood

**Kyra F. Mendes de Leon**[1,2,3,4]*, **Suzanne Jurriaans**[5], **Roos D. Klungers**[1,2,3,4], **Janke Schinkel**[5], **Pythia T. Nieuwkerk**[2,3,4], **Marc van der Valk**[1,3,6]

**1** Amsterdam UMC, University of Amsterdam, Infectious Diseases, Amsterdam, Netherlands,
**2** Amsterdam UMC, University of Amsterdam, Medical Psychology, Amsterdam, Netherlands,
**3** Amsterdam Institute for Immunology and Infectious Diseases (AII), Amsterdam, Netherlands,
**4** Amsterdam Public Health Research Institute, Amsterdam UMC, Amsterdam, Netherlands, **5** Amsterdam UMC, University of Amsterdam, Medical Microbiology and Infection Prevention, Amsterdam, Netherlands,
**6** Stichting hiv monitoring, Amsterdam, Netherlands

* k.f.mendesdeleon@amsterdamumc.nl

## Abstract

### Background

Home-based blood self-sampling for HIV-1 viral load (VL) monitoring has the potential to alleviate pressure on healthcare systems by reducing clinic visits for people with HIV. This study evaluates the feasibility and reliability of HIV-RNA measurements in self-sampled blood and in viremic samples.

### Methods

Between September 2024 and March 2025 participants were recruited at the outpatient clinic. Self-sampled blood was collected using the TassoPlus device at home and compared to conventional HIV-RNA testing from the same individual. Only samples containing a minimum volume of 200 µL plasma were deemed eligible for analysis. Also, 30 stored viremic samples (HIV-1 RNA 100–950 copies/mL) aliquoted into 200 µL volumes were included and compared to conventional HIV-RNA testing. The Alinity m HIV-1 assay (Abbott), validated for low plasma volumes, was used for all samples. Agreement was assessed using Pearson correlation and Bland-Altman analysis. Sample quality was assessed based on plasma volume and clotting. User-friendliness of the TassoPlus was evaluated through a questionnaire.

### Results

Of the 62 participants (median age 56 [46–66]; 81% male), 63% (n = 38) returned the sample, yielding a mean plasma volume of 207 µL (range:10–550). Of these, 18 samples (29%) were excluded due to insufficient volume for analysis. HIV-RNA in both self-sampled and stored samples (n = 50), correlated with conventional samples (r = 0.800). However, in four low-volume viremic samples, HIV-RNA was <100

**Data availability statement:** All relevant data are within the paper and its Supporting information files.

**Funding:** The author(s) received no specific funding for this work.

**Competing interests:** MvdV has received unrestricted research funding from ViiV, Gilead and MSD and fees for participation in scientific advisory boards from Viiv, Gilead Sciences and MSD all paid to his institution. KMdL, SJ, RK, JS and PN report no competing interests. This does not alter our adherence to PLOS ONE policies on sharing data and materials.

copies/mL, while conventional sampling detected 148, 260, 501 and 759 copies/mL, respectively. Bland-Altman analysis showed a mean difference of 0.08 log copies/mL (95%LOA:-0.79–0.95) between conventional sampled blood and low-volume samples (self-sampled/stored samples). No clotting was observed. Furthermore, 57% of participants expressed interest in using the TassoPlus, and 48% rated its usability as easy or very easy.

## Discussion

Self-sampling with TassoPlus presents challenges, including high rates of unsuccessful sampling and potential failure to detect low-level viremia. Therefore, significant refinements are essential for reliable clinical use.

---

## Introduction

The widespread implementation of antiretroviral therapy (ART) has fundamentally altered the clinical trajectory of HIV, enabling the majority of people with HIV to achieve sustained viral suppression [1]. Monitoring of plasma HIV-1 RNA levels, or viral load (VL), constitutes a critical component of the longitudinal assessment of ART effectiveness and treatment success [2]. In most international guidelines, VL monitoring is recommended on a biannual basis for individuals with stable virological control [3,4].

In the Netherlands, viral suppression rates are among the highest globally, with approximately 96% of individuals on ART achieving undetectable VL [5]. While this reflects the effectiveness of the national HIV care infrastructure, the healthcare system is facing increasing challenges [6]. This is partly due to the aging demographic of people with HIV, combined with limitations in healthcare resources [5,7]. Moreover, the growing utilization of healthcare services intensifies time constraints for healthcare professionals and poses barriers to delivering optimal care [7]. The increasing demands on the healthcare system emphasize the imperative for more efficient care delivery strategies.

Innovations in VL monitoring, such as remote or home-based alternatives, may offer a promising strategy to alleviate pressure on clinical services [8]. Remote sampling could reduce the number of in-person clinic visits while maintaining sufficient clinical oversight. For people with HIV, home-based viral load monitoring can help to overcome logistical barriers such as travel time and scheduling challenges while simultaneously enhancing convenience and autonomy [8,9]. From a healthcare system perspective, decentralizing routine monitoring may allow for more targeted use of clinical time and attention for individuals requiring intensified support [10].

TassoPlus is an innovative device that enables self-collection of capillary blood at home [11]. While it has been shown to perform well for other purposes, its use for HIV VL monitoring has not yet been evaluated [12–14]. In particular, it is unknown whether low-volume plasma samples, which are typically collected with this device, can provide reliable HIV-RNA measurements.

Our study aimed to assess the feasibility and reliability of HIV-RNA measurement in self-sampled blood in people with HIV using antiretroviral therapy (ART). Since low-level viremia in the Netherlands in people using ART is rare [15], we additionally assessed reliability of VL assessment in diluted low-volume stored viremic samples.

## Methods

### Study setting

A validation study was conducted between 9 September 2024 and 28 September 2025 at the Amsterdam University Medical Center (UMC), where care is provided to approximately 2,850 people with HIV.

Written informed consent was obtained from all participants. The study adhered to the principles of the Declaration of Helsinki and received approval from the Medical Ethics Review Committee (METC) of Amsterdam UMC (2023.0689).

Eligible participants were adults (≥18 years) with HIV, proficient in Dutch or English, and engaged in routine clinical HIV care. Two distinct groups were included: individuals on ART who were virally suppressed, and individuals with documented persistent low-level viremia (HIV-1 RNA 100–500 copies/mL).

Sample size was determined according to the method described by Lu for assessing agreement between two measurement methods using Bland-Altman analysis [15]. For each individual pair of viral load measurements (Tasso/stored samples versus conventional measurements), the difference was calculated, along with the mean difference and the 95% limits of agreement (LOA). The LOA represents the interval within which 95% of the differences between paired measurements are expected to lie. To assess whether the two methods were in sufficient agreement, a predefined clinically acceptable threshold was established. Method agreement was considered achieved if both the upper and lower calculated LOA did not exceed this threshold. We assumed a mean difference of 0.5 $\log_{10}$ cp/mL and a standard deviation of 0.4 $\log_{10}$ cp/mL. The clinically acceptable threshold was predefined as 2.0 $\log_{10}$ cp/mL. This margin respects the clinical decision thresholds whereby a HIV-1 RNA result of 200 cp/ml or higher in a person with previous suppressed HIV-1 RNA would trigger a discussion on adherence, drug-drug interactions and/or genotypic resistance test. Moreover, it accounts for the additional variability inherent in self-sampling. Based on these assumptions, a minimum of 12 participants was required to achieve 0.80 power, assuming a normal distribution. To account for the anticipated non-normal distribution of viral loads in the low-viremia subgroup, the sample size for this group was increased to 30, as the assumption of approximate normality is more reasonable with this sample size. A preliminary analysis estimated the number of people with HIV using ART exhibiting persistent low-level viremia. As this group alone was insufficient to fully assess viral load measurement with TassoPlus, 30 stored frozen viremic plasma samples with previously determined conventional VL (HIV-1 RNA 100–950 copies/mL) were aliquoted into 200 µL volumes, thus representative of the volume obtained by TassoPlus collection. Prior to testing, HIV negative plasma was added to the low volume samples to obtain the minimum required volume of 750 µL for viral load determination.

Participants unable to provide informed consent were excluded from the study.

### TassoPlus device

The TassoPlus (Tasso, Inc., Seattle, WA) device is a CE-marked capillary blood collection system designed for self-collection of capillary whole blood. During sample collection, the device is attached to the participant's upper arm. Activation by pressing a button deploys a retractable lancet, which punctures the skin, enabling blood to flow into the collection pod through a mild vacuum. The collection process typically takes approximately five minutes to obtain a maximum of 600 µL of blood. Upon completion, the device is detached and disposed of as a single-use medical device.

We prepared the TassoPlus kit, which included the single-use blood collection device, a separate collection tube with closure, printed instructions (available in Dutch and English) with a QR code linking to a demonstration video, in

accordance with the manufacturer's instructions for use [16]. In addition, the kit contained instructions for sample mailing, and a shipping envelope compliant with UN3373 and P650 regulations.

## Study procedure

Participants were recruited prior to their routine visits at the Amsterdam UMC outpatient clinic through the electronic patient portal. Healthcare providers identified eligible participants, either virally suppressed or with low-level viremia, and provided study information at least 24 hours before the scheduled appointment. During the appointment, their healthcare provider invited them to participate in the study. Those who consented were given additional written and verbal information and had the opportunity to discuss any questions with a researcher (KMdL) in a designated location within the outpatient clinic. After providing written informed consent, participants completed a brief questionnaire regarding demographic characteristics.

Participants were then provided with the TassoPlus kit along with written instructions, without any in-person demonstration. They underwent conventional viral load testing at the hospital and were asked to perform home self-sampling on the same day. The TassoPlus device was single-use and disposable. Participants collected the blood sample at home, recorded the date of collection on the return form, and sent it to the hospital using a prepaid envelope designed to fit in a standard mailbox. Upon receipt at the hospital, the plasma volume, clotting status, and postal transit time of the samples were recorded. Only samples containing a minimum of 200 µL of plasma were deemed eligible for analysis. Viral load testing was performed using the routine Alinity m HIV-1 assay (Abbott). To meet the minimum processing volume requirement of the analyzer (750 µL for aliquoted samples), all samples were manually diluted five-fold prior to testing. This procedure was performed in accordance with the manufacturer's instructions and conforms to FDA guidelines, [17] and has been validated previously in our lab. The Alinity m HIV-1 assay is a clinically validated, high-sensitivity RT-PCR assay, with established reliability for quantification of diluted, low-volume samples. Following the five-fold dilution, the effective lower limit of quantification was 100 copies/mL, maintaining the assay's capacity to detect low-level viremia. Stored low-viremic samples, previously assessed by conventional methods, were maintained at −70 °C until testing, underwent a single thaw, and were aliquoted into 200 µL volumes with a five-fold dilution to ensure consistency. One day after receiving the sampling kit, participants received a user-friendliness questionnaire by email through Castor EDC [18]. If participants failed to collect a blood sample independently, as indicated by the absence of blood in the collection tube during the initial attempt, a second collection was performed under the supervision of a researcher (RK) at the outpatient clinic following their routine appointment. During supervision, participants were asked to follow the same written instructions as in their previous attempt. The researcher observed the procedure and recorded any step at which the collection was unsuccessful without intervening. If blood flow was initiated, the time of onset and cessation was documented. This procedure aimed to determine whether an adequate sample could be obtained and to identify any procedural challenges.

No financial or other compensation was provided to participants for their contribution to the study.

## Statistical analysis

The primary outcome of this study was the reliability of self-collected Tasso Plus samples, evaluated by the agreement of HIV-1 RNA between self-sampled and conventional hospital-collected samples. Secondary outcomes included sample quality, feasibility of laboratory processing, and participant-reported usability and acceptability of the Tasso Plus device. Participant characteristics were summarized using descriptive statistics. Variables included age, gender, migration background (defined by the participant's country of birth or, if born in the Netherlands, the country of birth of their parents) [15], educational level (high, secondary, or primary/no education), employment status (paid or unpaid), household composition (living alone or with others), and travel time to the hospital (in minutes). Clinical characteristics included use of chronic daily oral medications other than HIV treatment (yes/no), years since HIV diagnosis, presumed route of HIV transmission, duration of antiretroviral therapy, and virological suppression (defined as HIV-1 viral load <200 copies/mL). Demographic

characteristics were compared between participants with successful and unsuccessful sampling. Categorical variables were compared using the chi-square test or Fisher's exact test, as appropriate. Continuous variables were assessed for normality using the Shapiro-Wilk test. Normally distributed variables were compared using independent samples t-tests, whereas non-normally distributed variables were compared using the Mann-Whitney U test.

## Feasibility

Feasibility was evaluated through sample quality, procedural performance, and user experience, with all outcomes analyzed using descriptive statistics. Sample quality was assessed by laboratory personnel based on plasma volume and evidence of clotting. Procedural performance was evaluated according to participant sampling outcomes. The overall return rate of TassoPlus kits was calculated as the number of tubes returned relative to the number distributed. Successful sampling was defined as returning a sample with measurable plasma, further categorized by whether the plasma volume was sufficient for analysis (≥200 μL) or insufficient (<200 μL). Unsuccessful sampling included participants who either attempted self-collection but did not obtain any blood or did not return a sample at all. For these participants, second supervised sampling attempts were conducted, and the time to blood onset and device detachment were recorded.

Postal transit time was calculated as the mean duration between TassoPlus sample collection and receipt at the hospital. The mean time interval between conventional venipuncture and TassoPlus self-sampling was also recorded.

User-friendliness was assessed using a post-sampling questionnaire administered via Castor EDC, with 5-point Likert scale responses summarized using descriptive statistics. All descriptive analyses were performed in SPSS version 28.

## Reliability

The reliability of 200 μL samples for viral load determination was evaluated based on the analytical performance of the self-collected and stored samples. This was determined by measuring the concordance between HIV-1 RNA results from 200 μL TassoPlus or stored viremic samples and conventional samples acquired by venipuncture. To ensure valid comparison, paired samples from TassoPlus and conventional venipuncture were first evaluated for the time interval between collection dates; pairs with intervals exceeding one week were excluded. Subsequently, HIV-1 RNA concentrations from 200 μL samples (both stored and TassoPlus) and conventional venipuncture samples were $\log_{10}$-transformed. Viral load values from 200 μL and conventional measurements were compared using Pearson's correlation. Agreement was further assessed using Bland-Altman analysis, plotting the differences between conventional and 200 μL measurements against their paired means. The mean difference (bias) and 95% limits of agreement (LOA) were subsequently calculated. All reliability analyses were conducted using SPSS version 28.

## Results

Among the 62 participants included in the study, 82% were male, with a median age of 56 years (IQR 46–66). Among participants who returned a sample containing plasma (n = 38), 72% had a higher education level, compared to 42% in the group with unsuccessful sampling or no sample returned despite follow-up attempts (n = 24; p = 0.031). Migration background differed significantly between participants with successful and unsuccessful sampling (p = 0.043). Participants from the Netherlands were more frequently represented in the successful group (70% vs. 55%), whereas participants from Latin America–Caribbean (8% vs. 25%) and Central and Southeast Asia (0% vs. 12%) were more frequent in the unsuccessful group (Table 1). Of the 62 participants included, 38 samples were returned, of which 22 contained sufficient plasma volume (≥200 μL) for analysis (Fig 1). The mean postal transit time was 3.3 days (SD: 5.8), and the mean time between conventional and TassoPlus blood collection was 1.5 days (SD: 7.0). No clotting was observed in any of the returned samples.

**Table 1. Characteristics of participants.**

| People with HIV | Total (N = 62) | | Successful blood sampling (n = 38) | | Unsuccessful blood sampling/ no return (n = 24) | | P-value |
|---|---|---|---|---|---|---|---|
| Age (years) | | | | | | | 0.430 |
| Median (IQR) | 56 | (46-66) | 60 | (45-67) | 53 | (46-64) | |
| Gender | | | | | | | 0.505 |
| Male | 51 | (82%) | 30 | (79%) | 21 | (88%) | |
| Female | 11 | (18%) | 8 | (21%) | 3 | (12%) | |
| Migration background | | | | | | | 0.043 |
| The Netherlands | 39 | (63%) | 26 | (70%) | 13 | (55%) | |
| Europe/North America/Australia | 6 | (10%) | 5 | (14%) | 1 | (4%) | |
| Latin America-Caribbean | 9 | (15%) | 3 | (8%) | 6 | (25%) | |
| Sub Saharan Africa | 4 | (6.5%) | 3 | (8%) | 1 | (4%) | |
| Central and Southeast Asia | 4 | (6.5%) | 0 | (0%) | 3 | (12%) | |
| Education | | | | | | | 0.031 |
| Primary or no education | 0 | (0%) | 0 | (0%) | 0 | (0%) | |
| Secondary | 24 | (40%) | 10 | (28%) | 14 | (58%) | |
| High | 36 | (60%) | 26 | (72%) | 10 | (42%) | |
| Labour status | | | | | | | 0.792 |
| Paid labour | 36 | (58%) | 23 | (60%) | 13 | (54%) | |
| No paid labour | 26 | (28%) | 15 | (40%) | 11 | (46%) | |
| Household composition | | | | | | | 0.958 |
| Alone | 23 | (25%) | 14 | (37%) | 9 | (38%) | |
| With others | 39 | (42%) | 24 | (63%) | 15 | (62%) | |
| Self-reported travel time to the hospital (minutes) | | | | | | | 0.386 |
| Median (IQR) | 30 | (20-45) | 30 | (19-45) | 30 | (20-60) | |
| Use of other chronic daily oral medication | | | | | | | 0.591 |
| Yes | 25 | (42%) | 23 | (38%) | 12 | (52%) | |
| No | 35 | (38%) | 14 | (62%) | 11 | (48%) | |
| Presumed HIV transmission route | | | | | | | 0.502 |
| MSM | 34 | (37%) | 22 | (58%) | 12 | (50%) | |
| Heterosexual | 15 | (16%) | 9 | (24%) | 6 | (25%) | |
| Vertical transmission | 1 | (1%) | 0 | (0%) | 1 | (4%) | |
| Unknown | 43 | (46%) | 7 | (18%) | 5 | (21%) | |
| Years on ART | | | | | | | 0.494 |
| Median (IQR) | 18 | (13-22) | 18 | (14-23) | 18 | (11-22) | |
| Years since HIV diagnosis | | | | | | | 0.682 |
| Median (IQR) | 22 | (18-27) | 22 | (18-26) | 21 | (18-29) | |
| HIV-1-RNA below 200 copies/ml | | | | | | | 0.518 |
| Yes | 60 | (97%) | 36 | (95%) | 24 | (100%) | |
| No | 2 | (3%) | 2 | (5%) | 0 | (0%) | |

Among the 38 self-collected samples that were returned with measurable plasma volume, the overall mean volume was 207 μL. When stratified by volume threshold, samples with sufficient volume (≥200 μL) had a mean plasma yield of 288 μL, whereas those with insufficient volume (<200 μL) had a mean yield of 99 μL.

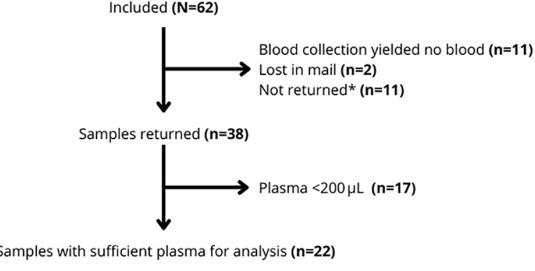

**Fig 1. Flowchart TassoPlus inclusion process.**

Of the 11 participants who were initially unsuccessful in collecting blood, nine agreed to a supervised follow-up attempt. Under supervision, all nine participants successfully obtained a blood sample. The mean plasma volume obtained during supervised sampling was 138 µL (range: 50–300 µL), with only one sample yielding ≥200 µL. The mean time to onset of blood collection was 3.9 minutes. The tube was filled and detached after a mean of 8.7 minutes, compared to the five minutes indicated in the manufacturer's written instructions.

The user-friendliness questionnaire was completed by 46 participants (Fig 2). Of these, 18 (39%) had a sample included in the final analysis (≥200 µL). Additionally, nine participants (20%) with unsuccessful blood collection completed the questionnaire, two participants did not perform the blood collection themselves, and one participant indicated not having read the paper-based instructions prior to collection. Seven participants (15%) indicated that the paper-based instructions did not adequately facilitate the blood collection process. More than half of participants (57%) expressed interest in using the TassoPlus device for future collections. When stratifying participants by blood collection success, 13% of those who did not succeed expressed interest in future use of the TassoPlus device, compared to 70% of those who successfully collected blood.

## Reliability of low-volume samples

Of the 22 samples with adequate plasma volume obtained with Tasso Plus, two were excluded from the reliability analysis due to an interval of 27 and 29 days with conventional sample collection. Therefore, 20 self-sampled and 30 stored viremic 200 µL samples were included in the reliability analysis and paired with their corresponding conventional viral load measurements. Among the conventional stored samples, the mean HIV-1 RNA was 413 copies/ml (range: 105–942 cp/ml).

Pearson correlation analysis showed a correlation of r = 0.800 between HIV-RNA levels in 200 µL and conventional samples. Bland-Altman analysis showed a mean difference of 0.08 log copies/mL (95%LOA: −0.79–0.95) between conventional sampled blood and 200 µL samples (self-sampled or stored samples). Four 200 µL samples fell outside the limits of agreement. In these cases, HIV-1 RNA levels in 200 µL samples were <100 copies/mL, while corresponding conventional samples yielded quantifiable viral loads 148, 260, 501 and 759 copies/mL, respectively (Fig 3). Furthermore, 18 of the 20 self-sampled 200 µL samples had undetectable HIV-1 RNA levels, appearing as a single overlapping point in Fig. 3.

## Discussion

Self-sampling represents an innovative approach to decentralize HIV-RNA monitoring, offering the potential to reduce the burden on healthcare systems and enhance autonomy in care for people with HIV. To the best of our knowledge, this is the first study to evaluate the use of the TassoPlus device for HIV-RNA testing with low-volume samples using the Alinity m HIV-1 assay. We show that, in its current form, this method is not feasible for routine clinical use, primarily due to

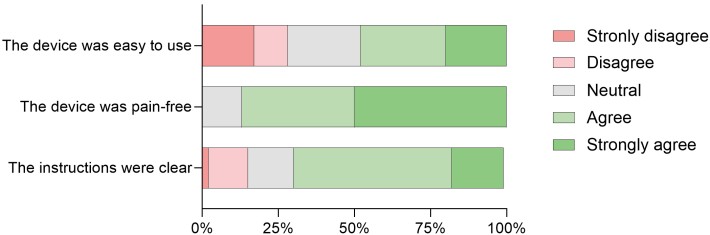

**Fig 2. Response distribution across Likert-scale items (N=46).**

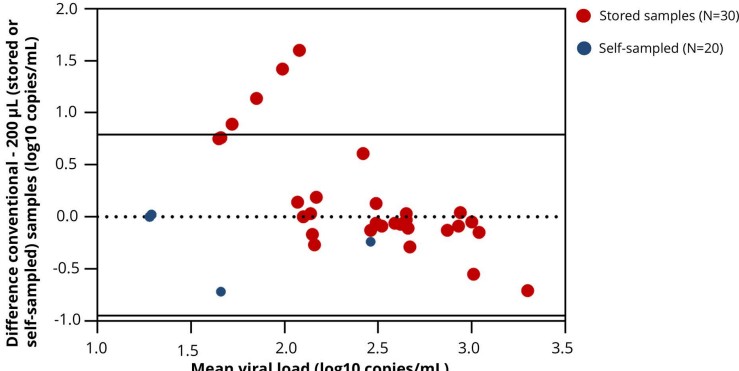

**Fig 3. Bland-Altman plot comparing HIV-RNA levels in conventional samples versus 200 μL samples (stored or self-collected using TassoPlus).**

the high rate of unsuccessful blood collections (53%) and the risk of failing to detect low-level viremia in 200 μL samples (13%).

Several studies have evaluated the TassoPlus device for other clinical applications. For example, a study evaluating TassoPlus for COVID-19 antigen detection reported an 80% first-attempt success rate for blood collection, with a mean plasma volume of 287 μL, which is notably higher than the mean volume observed in our study (207 μL) [14]. However, their study procedure included supervision during the initial attempt, and the authors reported that participants collecting blood unsupervised were more prone to mistakes [14]. Our results similarly indicate that supervision increases the likelihood of successful blood collection, although it does not necessarily ensure sufficient plasma volume, possibly due to procedural factors or the intrinsic performance characteristics of the device. In addition, variability in reported outcomes may reflect differences in the definition of "successful" sampling specific to the TassoPlus device. In this study, success was defined as returning a sample with ≥200 μL of plasma, which is stricter than in some previous studies. For example, one study reported a 70% success rate using a threshold of ≥80 μL serum, illustrating how differences in volume criteria can affect reported success rates [19].

Plasma is the preferred specimen for sensitive and reliable HIV-1 RNA quantification [20]. The ability to detect low-level viremia from limited plasma volumes, however, has not been well characterized before. When tested using the Alinity m HIV-1 assay (Abbott) according to the manufacturer's protocol [17], a plasma input volume of 200 μL was insufficient for robust viral load determination in low-viremic samples, resulting in missed detection of low-level viremia in 13% of cases. Although a single low-level viremia measurement is usually clinically insignificant, it becomes relevant when persistence goes undetected, as a subsequent measurement would typically follow and may likewise be missed when using

low-volume testing. This limitation may delay the recognition of ongoing viral replication, which over time could contribute to virologic failure and an increased risk of HIV transmission [21]. While the smaller plasma volumes obtained with devices such as TassoPlus may be sufficient for antigen detection, serology or other diagnostic applications, as evidenced in previous studies, they are suboptimal for nucleic acid-based assays, which require larger inputs to achieve high analytical sensitivity [13,19]. One study evaluating the TassoPlus device for capillary blood self-sampling assessed plasma cytomegalovirus DNA load using the Abbott M2000rt qPCR assay. In that study, TassoPlus collection yielded a mean of 590 µL of capillary blood, corresponding to approximately 325–355 µL of plasma. The smaller plasma volume limited the assay input and resulted in reduced analytical sensitivity, approximately tenfold lower than that of standard venous plasma tested with the Abbott M2000rt assay. Viral loads measured in TassoPlus samples were generally comparable to those in venous plasma, but reliable detection and quantification of CMV DNA in TassoPlus samples required higher viral loads in venous plasma [12]. These observations demonstrate that assay sensitivity is closely dependent on plasma volume. Overall, assay sensitivity improves with increasing plasma volume [22] yet the precise minimum threshold required for accurate detection of low-level viremia remains to be established.

In addition to these analytical limitations, low-volume sampling introduces practical laboratory challenges. The Abbott Alinity m HIV-1 analyzer used in our study requires a minimum of 750 µL of plasma; consequently, all 200 µL samples had to be diluted prior to testing [17]. This additional step is more labor-intensive for laboratory personnel compared with conventional viral load testing, where samples can be directly loaded onto the analyzer. Furthermore, the limited plasma volume obtained from TassoPlus collections precludes the availability of backup material for repeat testing in the event of technical failure or other issues.

The feasibility of self-sampling was limited by practical difficulties. Eleven of 50 self-performed blood collections (22%) failed, and 11 of 62 samples (18%) were not returned despite follow-up efforts, representing both a loss of resources and a potential risk of failing to monitor individuals who may require clinical follow-up. Participants' perceptions of the device may partly explain these outcomes. While most participants agreed that the procedure was pain-free and that the instructions were clear, perceptions of usability were more variable, with only 48% finding the device easy to use. Exploratory analyses indicated that participants who successfully returned and completed sampling tended to have higher educational levels, although the sample size was insufficient to draw definitive conclusions. Overall, 57% of participants expressed interest in using the TassoPlus device, increasing to 70% among those who successfully collected a sample. In prior studies, higher levels of usability and interest were reported, which may reflect the provision of more extensive guidance or supervision during sample collection [13,19]. Taken together, these findings indicate that with targeted support strategies or improved user instructions, return rates and overall feasibility of self-sampling could be improved.

Despite existing challenges, future advances could enable self-sampling to offer a cost-saving strategy for monitoring HIV-1 RNA. With increasing pressure on healthcare budgets, cost-reducing strategies are particularly important. Current guidelines recommend biannual outpatient consultations with standard laboratory testing [3,4]. Implementing self-sampling could provide a personalized approach, whereby one routine consultation and associated laboratory testing may be substituted by the cost of a self-sampling device and laboratory analysis, resulting in lower overall healthcare costs.

Several limitations of this study should be acknowledged. First, participants performed self-sampling with only written instructions and without in-person supervision, which may have influenced both sample collection success and user experience. Because self-collection was conducted at home, we were unable to assess the specific steps at which mistakes occurred or whether participants fully adhered to the instructions, which may have contributed to the relatively high rate of failed collections. The advantage of the current approach is that it reflects real-life circumstances. However, providing participants with prior practical training on the device in future studies may improve success rates, as demonstrated by previous research [14]. Second, a substantial proportion of samples were derived from stored plasma that had been previously frozen, thawed, and subsequently diluted to meet the minimum input volume required for the Abbott Alinity m HIV-1 assay (750 µL). However, this 5-fold dilution corresponds to an effective lower limit of quantification of approximately 100

copies/mL, which in theory should have been sufficient to detect viral loads in our samples (100−950 copies/mL). Nonetheless, this approach likely reduced assay sensitivity and may have introduced bias when comparing viral load measurements to conventional testing using undiluted samples with lower limit of quantification. Additionally, the freeze-thaw process itself could have influenced viral RNA stability and, consequently, viral load measurements. However, samples were only thawed once, which is generally not expected to compromise HIV RNA stability [23]. Finally, we were only able to recruit two participants with low-level viremia, of whom only one met the criteria to be included in the analysis of reliability of the TassoPlus device. This limits both the reliability assessment and the generalizability of findings for this group, as the remaining low-level viremic samples were derived from previously stored plasma rather than self-collected samples. Future studies should focus on strategies to improve the success of self-sampling. First, it is important to establish the minimum plasma volume required for reliable detection of low-level HIV-1 RNA, which will guide the selection of self-sampling devices capable of consistently providing sufficient plasma for accurate and sensitive viral load measurement. Additionally, providing participants with practical training on device use may further increase sampling success. These improvements could facilitate the development of future self-sampling approaches for accurate HIV-1 RNA monitoring. In its current form, 200 µL self-collection using the TassoPlus device is not feasible for reliable detection of HIV-1 RNA in diluted samples using the Abbott Alinity m HIV-1 assay, particularly for low-level viremia. Nonetheless, participants showed considerable interest in home-based self-collection, highlighting the potential for innovation and optimized procedures to make self-sampling a feasible and reliable option for selected individuals in the future.

## Supporting information

**S1 File. Dataset for agreement between low-volume and conventional methods.** https://doi.org/10.6084/m9.figshare.32112460.
(SAV)

## Acknowledgments

We thank the participants for their contributions to the study.

## Author contributions

**Conceptualization:** Kyra F. Mendes de Leon, Suzanne Jurriaans, Janke Schinkel, Pythia T. Nieuwkerk, Marc van der Valk.

**Data curation:** Kyra F. Mendes de Leon, Pythia T. Nieuwkerk.

**Formal analysis:** Kyra F. Mendes de Leon, Pythia T. Nieuwkerk.

**Methodology:** Kyra F. Mendes de Leon, Suzanne Jurriaans, Pythia T. Nieuwkerk, Marc van der Valk.

**Resources:** Suzanne Jurriaans.

**Supervision:** Pythia T. Nieuwkerk, Marc van der Valk.

**Writing – original draft:** Kyra F. Mendes de Leon.

**Writing – review & editing:** Kyra F. Mendes de Leon, Suzanne Jurriaans, Roos D. Klungers, Janke Schinkel, Pythia T. Nieuwkerk, Marc van der Valk.

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
