## [Decision Letter · Decision Letter 0]

3 Mar 2026

PONE-D-25-65444Feasibility and reliability of decentralized HIV-1 viral load monitoring on self-sampled bloodPLOS One

Dear Dr. Mendes de Leon,

Thank you for submitting your manuscript to PLOS ONE. After careful consideration, we feel that it has merit but does not fully meet PLOS ONE’s publication criteria as it currently stands. Therefore, we invite you to submit a revised version of the manuscript that addresses the points raised during the review process.

Your manuscript was reviewed by two experts in the field. Both identified many important issues in your submission that require your careful attention. Please thoroughly review the attached comments and provide point-by-point responses.

We look forward to receiving your revised manuscript.

Kind regards,

Yury E Khudyakov, PhD

Academic Editor

PLOS One

Journal Requirements:

“MvdV has received unrestricted research funding from ViiV, Gilead and MSD and fees for participation in scientific advisory boards from Viiv, Gilead Sciences and MSD all paid to his institution. KMdL, SJ, RK, JS and PN report no competing interests.”

We note that one or more of the authors are employed by a commercial company: name of commercial company.

provide an amended Funding Statement declaring this commercial affiliation, as well as a statement regarding the Role of Funders in your study. If the funding organization did not play a role in the st udy design, data collection and analysis, decision to publish, or preparation of the ma

a.        Please nuscript and only provided financial support in the form of authors' salaries and/or research materials, please review your statements relating to the author contributions, and ensure you have specifically and accurately indicated the role(s) that these authors had in your study. You can update author roles in the Author Contributions section of the online submission form.

Reviewers' comments:

Reviewer's Responses to Questions

**Comments to the Author**

1. Is the manuscript technically sound, and do the data support the conclusions?

Reviewer #1: Partly

Reviewer #2: Partly

2. Has the statistical analysis been performed appropriately and rigorously? 

Reviewer #1: No

Reviewer #2: Yes

3. Have the authors made all data underlying the findings in their manuscript fully available?

Reviewer #1: No

Reviewer #2: Yes

4. Is the manuscript presented in an intelligible fashion and written in standard English?

Reviewer #1: Yes

Reviewer #2: Yes

5. Review Comments to the Author

Reviewer #1: This study sought to assess the feasibility and reliability of home-based HIV-1 viral load (VL) monitoring using the TassoPlus self-sampling device, by comparing results from self-collected dried blood and viramic samples with conventional clinic-based HIV-RNA testing. The authors also evaluated user-friendliness of the device via a questionnaire. After a thorough review, I believe there are additional issues that need to be addressed as follows:

1. Although this study, which evaluates the application of the TassoPlus device for HIV VL monitoring, has certain innovativeness, it suffers from insufficient sample size and lack of methodological rigor, resulting in inadequate strength of research evidence.

2. The study only conducted a single sampling assessment and failed to supplement key dimensions such as long-term adherence and applicability to different populations. In addition, the sample collection success rate is low, with experimental quality defects, making it impossible to fully verify the feasibility of the device.

3. The citation of references is non-standard.

4. Although the authors claim that "All relevant data are within the manuscript and its Supporting Information files", I have not seen the relevant data uploaded by the authors.

Reviewer #2: It is generally a well-conceived study with relevant question.

Below are a few comments:

1. In methodology, it could be nice to clearly spell what the primary and secondary outcomes are. I do note the aims are feasibility and reliability- but clearly noting what the primary outcome under each would help in making conclusions.

2.Regarding sample size calculation- do you have reference to support the assumptions made particulary on +/- 2.0 log? if so might be worth supporting with literature.

3. Could you provide a stronger rationale for choosing the alinity m platform despite its 750ul input requirement or discuss whether platforms requiring smaller inputs could alter results

4. Clearly describe the recruitment strategy. For example how did you identify the virally suppressed vs low level viremia participants- was it consecutive?

5. Regarding supervised sampling- what really happened during supervision? another aspect to admit whether in discussion is whether incorporation of prior training would make a difference.

6. Analysis of feasibility and usability could be strengthened by exploratory comparisions eg education level vs success using appropriate tests, with p values.

7. In the results, the text narrative could be improved by taking the reader through each attrition step with numbers and percentages. The current text is a little fragmented in describing fig 1.

8. Figure 3 is alittle confusing- it shows only few black dots compared to the number indicated in the text.

9. In Discussion, i found some claims/points contradictin . For example you do discuss the rationale of choosing acceptable sample of 200ul in this study which is specific to the objectives. There is no point in saying that success would have improved if low sample volume was choosen. In saying that, the objectives and methodology could not have been sensible for this particular study. You may wish to refraim your statements.

10. The conclusion made in this study should be carefuly framed to align within the limitation of specific device assay and protocol used. Might be worth suggesting what can be done reasonably better to improve succes.

11. Last paragraph regarding future work could be more actionable. Suggestions would be to use altenative testing that yield more for example, incorporating physical trainings to participants. Might also worth discussing briefly cost related implication

6. PLOS authors have the option to publish the peer review history of their article (what does this mean?). If published, this will include your full peer review and any attached files.

Reviewer #1: No

Reviewer #2: No

---

## [Author Response · Author response to Decision Letter 1]

14 Apr 2026

Dear Sir/Madam,

Please find the revised manuscript entitled, “Feasibility and reliability of decentralized HIV-1 viral load monitoring on self-sampled blood” for publication as an original article in PLOS ONE.

We appreciate the time that the editors and reviewer have taken to review our manuscript and the constructive comments provided. We have modified the manuscript accordingly. Changes to the text, tables and footnotes of the tables in the manuscript are also marked in tracked changes.

We have also provided a version without marked changes. All authors have seen and approved this version of the manuscript.

Thank you again for considering our revised manuscript and we sincerely hope it is now ready for publication in PLOS ONE.

Sincerely,

on behalf of the authors,

Kyra Mendes de Leon

Reviewers comments

Reviewer #1: This study sought to assess the feasibility and reliability of home-based HIV-1 viral load (VL) monitoring using the TassoPlus self-sampling device, by comparing results from self-collected dried blood and viramic samples with conventional clinic-based HIV-RNA testing. The authors also evaluated user-friendliness of the device via a questionnaire. After a thorough review, I believe there are additional issues that need to be addressed as follows:

Answer:

We thank the reviewer for their careful review and valuable feedback on our study. Our responses to the comments are provided below.

1. Although this study, which evaluates the application of the TassoPlus device for HIV VL monitoring, has certain innovativeness, it suffers from insufficient sample size and lack of methodological rigor, resulting in inadequate strength of research evidence.

Answer:

Thank you for highlighting this. The sample size was determined a priori to determine agreement between the two measurement methods. To achieve a statistical power of 0.80, a minimum of 12 participants was required in the suppressed viral load group and 30 in the low-level viremia group. After excluding samples with insufficient volume for detection, 18 people with HIV with suppressed viral loads were included, which, in our opinion, is still sufficient to address our research question based on the power calculation. Low-level viremia is relatively rare, making it difficult to recruit enough eligible participants. To overcome this limitation, we supplemented the study with prospectively collected stored samples, adding 30 low-viremic samples to the two participants initially enrolled, resulting in a total of 32 low-viremic samples. This sample size is adequate to assess agreement between the methods.

Regarding the reviewer’s comment on “lack of methodological rigor,” it is difficult to address this point directly because it is unclear whether the concern relates to sample size or to other aspects of the study methodology. We believe that the methods employed are well-suited to address our research objectives, specifically to assess whether HIV-RNA measurements are consistent when using low-volume samples compared to the conventional measurements, and we evaluated this using Bland-Altman analysis. This is a widely accepted and robust method for assessing agreement between two quantitative measurement techniques, providing a clear and reliable evaluation of method comparability. We previously used the same method to assess HCV-RNA on dry blood spots https://doi.org/10.1371/journal.pone.0231385 .

2. The study only conducted a single sampling assessment and failed to supplement key dimensions such as long-term adherence and applicability to different populations. In addition, the sample collection success rate is low, with experimental quality defects, making it impossible to fully verify the feasibility of the device.

Answer:

We thank the reviewer for raising this important point. The low sample collection success rate was an explicit part of our pre-defined study objectives, as the aim was to assess the feasibility of the method when participants collect samples themselves. Measuring the success rate under these conditions therefore, in our opinion, directly contributes to the assessment of feasibility of the device.

With respect to long-term adherence and applicability to other populations, these aspects were beyond the scope of the present study. Based on our findings, the use of the device to self-sample HIV-1 RNA seems not feasible in this population. If the method were found to be feasible under different conditions, future studies could then appropriately explore adherence and performance across diverse populations.

3. The citation of references is non-standard.

Answer:

We would like to clarify that the references are cited in Vancouver style, in accordance with the submission guidelines of PLOS ONE.

4. Although the authors claim that "All relevant data are within the manuscript and its Supporting Information files", I have not seen the relevant data uploaded by the authors.

Answer:

We thank the reviewer for this comment. Given the small sample size, including demographic characteristics could risk participant identifiability, which is why the full dataset was not previously included. Individual data points can still be seen in the Bland-Altman plots. To address this point by the reviewer, we have now uploaded the dataset without any demographic information, while ensuring that all relevant data necessary to reproduce the results are included.

Reviewer #2: It is generally a well-conceived study with relevant question.

Below are a few comments:

Answer:

Thank you for your positive feedback and for recognizing the relevance of our study. We appreciate your comments on our methodology and the limitations of our work. Our responses to the comments are provided below.

1. In methodology, it could be nice to clearly spell what the primary and secondary outcomes are. I do note the aims are feasibility and reliability- but clearly noting what the primary outcome under each would help in making conclusions.

Answer:

Thank you for this suggestion, we have accordingly adjusted the Methods section, line 225, page 10:

The primary outcome of this study was the reliability of self-collected Tasso Plus samples, evaluated by the agreement of HIV-1 RNA between self-sampled and conventional hospital-collected samples. Secondary outcomes included sample quality, feasibility of laboratory processing, and participant-reported usability and acceptability of the Tasso Plus device.

2.Regarding sample size calculation- do you have reference to support the assumptions made particulary on +/- 2.0 log? if so might be worth supporting with literature.

Answer:

We would like to clarify the rationale for the chosen margin of ±2.0 log₁₀ copies/ml, which was a trade-off between analytical accuracy and practical feasibility. If the self-sampling method performed within these margins, it could be used for clinical monitoring of people with a well-treated HIV-infection. This margin respects the clinical decision thresholds whereby a HIV-1 RNA result of 200 c/ml or higher in a person with previous suppressed HIV-1 RNA would trigger a discussion on adherence, drug-drug interactions and or genotypic resistance test. Moreover, it accounts for the additional variability inherent in self-sampling.

To improve clarity, we have added a more detailed explanation in the Methods section (Original text, Methods, line 150, page 6):

We assumed a mean difference of 0.5 log₁₀ cp/mL and a standard deviation of 0.4 log₁₀ cp/mL. The clinically acceptable threshold was predefined as 2.0 log₁₀ cp/mL. Based on these assumptions, a minimum of 12 participants was required to achieve 0.80 power, assuming a normal distribution. To account for the anticipated non-normal distribution of viral loads in the low-viremia subgroup, the sample size for this group was increased to 30, as the assumption of approximate normality is more reasonable with this sample size.

Revised text:

We assumed a mean difference of 0.5 log₁₀ cp/mL and a standard deviation of 0.4 log₁₀ cp/mL. The clinically acceptable threshold was predefined as 2.0 log₁₀ cp/mL. This margin respects the clinical decision thresholds whereby a HIV-1 RNA result of 200 c/ml or higher in a person with previous suppressed HIV-1 RNA would trigger a discussion on adherence, drug-drug interactions and or genotypic resistance test. Moreover, it accounts for the additional variability inherent in self-sampling. Based on these assumptions, a minimum of 12 participants was required to achieve 0.80 power, assuming a normal distribution. To account for the anticipated non-normal distribution of viral loads in the low-viremia subgroup, the sample size for this group was increased to 30, as the assumption of approximate normality is more reasonable with this sample size.

3. Could you provide a stronger rationale for choosing the alinity m platform despite its 750ul input requirement or discuss whether platforms requiring smaller inputs could alter results

Answer:

Thank you for this important comment. The Abbott Alinity m HIV-1 assay was selected because it is the routine clinical platform used in our laboratory for HIV-1 viral load monitoring and provides highly sensitive and reliable quantification. Although the analyzer requires a minimum processing volume of 750 µL for aliquoted samples, the platform is validated for accurate testing of diluted samples according to the manufacturer’s protocol.

In our study, samples were diluted five-fold prior to testing. Even with this dilution factor, the resulting lower limit of quantification remained highly sensitive, and is able to detect viral loads > 100 copies/mL, which remains suitable for detecting low-level viremia. Importantly, testing of five-fold diluted samples is routinely applied in our lab, after internal validation, when samples obtained via standard blood draws yield insufficient EDTA plasma. This occurs, for example, in neonates born to mothers with HIV, children with HIV, or people who are difficult to phlebotomize. Many RT-PCR-based viral load assays require comparable processing volumes. Platforms accepting substantially smaller input volumes are often point-of-care systems, which typically have higher limits of quantification or different analytical performance characteristics. As a result, the use of alternative platforms could potentially alter results, although this would also depend on multiple analytical and procedural factors and was therefore beyond the scope of the present study. However, we have clarified our rationale for choosing the Alinity m platform more explicitly in the revised text (see below).

Original text, Methods, line 199, page 9:

Viral load testing was performed using the routine Alinity m HIV-1 assay (Abbott). To meet the minimum processing volume requirement of the analyzer (750 µL for aliquoted samples), all samples were manually diluted five-fold prior to testing. This procedure was performed in accordance with the manufacturer’s instructions and conforms to FDA guidelines [17].

Revised text:

Viral load testing was performed using the routine Alinity m HIV-1 assay (Abbott). To meet the minimum processing volume requirement of the analyzer (750 µL for aliquoted samples), all samples were manually diluted five-fold prior to testing. This procedure was performed in accordance with the manufacturer’s instructions and conforms to FDA guidelines [17] and has been validated previously in our lab. The Alinity m HIV-1 assay is a clinically validated, high-sensitivity RT-PCR assay, with established reliability for quantification of diluted, low-volume samples. Following the five-fold dilution, the effective lower limit of quantification was 100 copies/mL, maintaining the assay’s capacity to detect low-level viremia.

4. Clearly describe the recruitment strategy. For example how did you identify the virally suppressed vs low level viremia participants- was it consecutive?

Answer:

Thank you for your comment. We agree that this point was not entirely clear in the original manuscript and have therefore clarified it in the text, as shown below.

Original text, Methods, line 182, page 8:

Participants received study information at least 24 hours prior to their scheduled routine clinical appointment through the electronic patient portal. During the appointment, their healthcare provider invited them to participate in the study. Those who consented were given additional written and verbal information and had the opportunity to discuss any questions with a researcher (KMdL) in a designated location within the outpatient clinic. After providing written informed consent, participants completed a brief questionnaire regarding demographic characteristics.

Revised text:

Participants were recruited prior to their routine visits at the Amsterdam UMC outpatient clinic through the electronic patient portal. Healthcare providers identified eligible participants, either virally suppressed or with low-level viremia, and provided study information at least 24 hours before the scheduled appointment. During the appointment, their healthcare provider invited them to participate in the study. Those who consented were given additional written and verbal information and had the opportunity to discuss any questions with a researcher (KMdL) in a designated location within the outpatient clinic. After providing written informed consent, participants completed a brief questionnaire regarding demographic characteristics.

5. Regarding supervised sampling- what really happened during supervision? another aspect to admit whether in discussion is whether incorporation of prior training would make a difference.

Answer:

Thank you for this question. We have further elaborated on this point in both the Methods and Discussion sections as shown below.

Original text, Methods, line 211, page 9:

If participants failed to collect a blood sample independently, as indicated by the absence of blood in the collection tube during the initial attempt, a second collection was performed under the supervision of a researcher (RK) at the outpatient clinic following their routine appointment. For cases in which the collection was unsuccessful, study personnel recorded the step at which the process failed.

Revised text:

If participants failed to collect a blood sample independently, as indicated by the absence of blood in the collection tube during the initial attempt, a second collection was performed under the supervision of a researcher (RK) at the outpatient clinic following their routine appointment. During supervision, participants were asked to follow the same written instructions as in their previous attempt. The researcher observed the procedure and recorded any step at which the collection was unsuccessful without intervening.

Original text, Discussion, line 441, page 20:

Several limitations of this study should be acknowledged. First, participants performed self-sampling with only written instructions and without in-person supervision, which may have influenced both sample collection success and user experience. Because self-collection was conducted at home, we were unable to assess the specific steps at which mistakes occurred or whether participants fully adhered to the instructions, which may have contributed to the relatively high rate of failed collections. The advantage of the current approach is that it reflects real-life circumstances.

Revised text:

Several limitations of this study should be acknowledged. First, participants performed self-sampling with only written instructions and without in-person supervision, which may have influenced both sample collection success and user experience. Because self-collection was conducted at home, we were unable to assess the specific steps at which mistakes occurred or whether participants fully adhered to the instructions, which may have contributed to the relatively high rate of failed collections. The advantage of the current approach is that it reflects real-life circumstances. However, providing participants with prior practical training on the device in future studies may improve success rates, as demonstrated by previous research [14].

6. Analysis of feasibility and usab

---

## [Editor Report · Decision Letter 1]

20 Apr 2026

Feasibility and reliability of decentralized HIV-1 viral load monitoring on self-sampled blood

PONE-D-25-65444R1

Dear Dr. Mendes de Leon,

We’re pleased to inform you that your manuscript has been judged scientifically suitable for publication and will be formally accepted for publication once it meets all outstanding technical requirements.

Kind regards,

Yury E Khudyakov, PhD

Academic Editor

PLOS One
---

## [Editor Report · Acceptance letter]

PONE-D-25-65444R1

PLOS One

Dear Dr. Mendes de Leon,

I'm pleased to inform you that your manuscript has been deemed suitable for publication in PLOS One. Congratulations! Your manuscript is now being handed over to our production team.

Kind regards,

on behalf of

Dr. Yury E Khudyakov

Academic Editor

PLOS One